# PARALLEL PROMPTING: FAST LLM INFERENCE FOR SHARED-CONTEXT, SHORT-TO-MODERATE OUTPUT

## ABSTRACT

We introduce *Parallel Prompting*, a method for high-throughput, quality-preserving decoding of multiple large language model (LLM) queries that share a common prefix. Such shared-context structure arises naturally in applications including document question answering, few-shot learning, multi-user chat, and evaluation pipelines. Prior approaches either degrade generation quality by merging queries into a single prompt that the model cannot reliably disentangle or impose rigid batching and preallocated memory that limit practical deployment. Parallel Prompting is a free lunch for batch prompting: it improves throughput and memory efficiency without requiring model retraining or sacrificing accuracy. The gains are most pronounced when prefix overlap is high and output lengths are short to moderate, with the relative advantage diminishing as unique suffixes grow longer.

Our method executes a single pass over the shared context and decodes all continuations in parallel through efficient matrix–matrix operations, while avoiding cross-query interference and supporting flexible batching across multiple sharing groups with dynamic, on-demand KV-cache management. This design enables high resource utilization during decoding without compromising output quality. Experiments on popular datasets with Llama 3-8B show up to a 4× reduction in end-to-end latency relative to competitive baselines, with no loss in accuracy, demonstrating that Parallel Prompting complements existing batching strategies and expands the practical throughput of LLM-based systems.

## 1 INTRODUCTION

Batch text generation is a standard paradigm for large language model (LLM) inference. In many practical scenarios, prompts within a batch often share a common prefix. This setting is prevalent in wide range of use-cases, such as document question answering, few-shot learning, multi-user chat, LLM-as-judges for model evaluation, and LLM-based verification for fact-checking. For instance, chatbots frequently serve diverse users using a shared system prompt, assistant models leverage few-shot exemplars for domain-specific tasks, and programming systems generate multiple candidate solutions to a single problem. As deployment of transformer-based LLMs continues to scale, harnessing these shared prefixes for efficiency becomes increasingly valuable.

A growing body of work seeks to accelerate LLM inference by exploiting shared information across requests. Several systems (Zhu et al., 2024; Juravsky et al., 2024) reuse parts of the cache when different prompts begin with the same prefix, thereby avoiding redundant computation. While these approaches achieve meaningful speedups, they remain limited in important ways: some require rigid memory layouts, and others only handle batches in which all inputs share exactly the same prompt. Related methods (Kwon et al., 2023; Zheng et al., 2024; Gim et al., 2024) extend cache reuse further but still follow a fundamentally sequential decoding pattern, leaving substantial efficiency gains unrealized. Meanwhile, simple batch prompting strategies that merge multiple queries into a single prompt often degrade output quality because the model cannot reliably separate the different requests. These limitations highlight the need for a method that simultaneously avoids interference, supports flexible sharing groups, and fully exploits parallelism during decoding.

In this paper, we propose **Parallel Prompting**, a method for efficiently decoding multiple queries with a shared prefix by processing them in parallel. The key insight is that we can independently encode each query with respect to the shared context using specialized attention masks, then generate outputs

in parallel during the decoding phase. This approach leverages efficient matrix-matrix operations on modern GPUs to achieve significant speedups without compromising output quality. Critically, we find that maximizing throughput requires carefully balancing two parameters: the batch size and the degree of parallelism during decoding—the optimal point depends on hardware and model specifics.

To summarize, our work makes the following contributions:

- We propose a simple and effective method leveraging parallel prompting in LLM that allows efficient batching of multiple LLM prompts which share a prefix.
- We conduct extensive experiments and show that our method can achieve improvements in throughput and computational resource management over prior methods across a range of workloads, although there are some workloads for which our proposed method is less efficient than some prior methods.
- We show theoretically and experimentally that maximizing inference throughput for parallel prompting requires a careful balance between attention parallelism and batch size.

| Prompt |
| --- |
| Saturn's inside is similar to Jupiter's inside. At the centre is a small rocky core that is about the size of the Earth. Saturn's core is very hot. |
| Q1: Saturn's inside is similar to what planet? |
| Q2: What is at the center of Saturn? |
| Q3: What temperature of Saturn's core? |

| | Generation | | | | | |
| --- | --- | --- | --- | --- | --- | --- |
| Generation Timestep | Batch Prompting (Cheng et al 2023) | | | Our Method | | |
| t=0 | A1: | | | A1: | A2: | A3: |
| t=1 | A1: Jupiter | | | A1: Jupiter | A2: A | A3: very |
| t=2 | A1: Jupiter | A2: | | A1: Jupiter | A2: A small | A3: very hot |
| t=3 | A1: Jupiter | A2: A | | A1: Jupiter | A2: A small rocky | A3: very hot |
| t=4 | A1: Jupiter | A2: A small | | | | |
| t=5 | A1: Jupiter | A2: A small rocky | | | | |
| t=6 | A1: Jupiter | A2: A small rocky core | | | | |
| t=7 | A1: Jupiter | A2: A small rocky core | A3: | | | |
| t=8 | A1: Jupiter | A2: A small rocky core | A3: very | | | |
| t=9 | A1: Jupiter | A2: A small rocky core | A3: very hot | | | |

Figure 1: Overview of our method. The input is a prompt with a shared context and multiple questions. Batch prompting (Cheng et al., 2023; Lin et al., 2024) concatenates all questions together, and the output is generated squentially using the typical LLM decoding method, taking 9 generation timesteps. Our method generates the output in parallel and produces the result faster, taking only 3 generation timesteps.

Our approach is a free lunch for batch prompting: it boosts throughput and memory efficiency without requiring any model retraining and without compromising accuracy. The gains are largest when prefix overlap is high and outputs are short to moderate, with the relative advantage tapering off as unique suffixes grow longer. Unlike simple batch-prompting heuristics—which often degrade generation quality by forcing the model to disentangle multiple requests within a single prompt—our method avoids cross-query interference, supports flexible sharing groups, and fully exploits parallelism throughout decoding.

## 2  BACKGROUND: ATTENTION MECHANISM

A core component of the Transformer is the attention computation. Given the sequence of queries $Q \in \mathbb{R}^{N_q \times d}$, keys $K \in \mathbb{R}^{N_{kv} \times d}$, values $V \in \mathbb{R}^{N_{kv} \times d}$, the transformer model computes the attention output $O \in \mathbb{R}^{N_q \times d}$ with the causal masking $M$ as follows:

$$O = \text{Attention}(Q, K, V) = \text{softmax}\left(\frac{QK^T}{\sqrt{d}} + M\right)V \tag{1}$$

At the start of the generation process, a prefill stage processes the initial sequence of tokens that the LLM will complete. During this stage, the entire prompt is encoded in parallel using a single transformer forward pass. This results in a high number of queries and key-value pairs ($N_q = N_{kv} \gg 1$), making the matrix multiplications in Equation 1 more hardware-friendly.

As the generation continues, completion tokens are decoded sequentially, with each decoding step producing a new token and requiring a forward pass. To speed up this process, a KV cache is used to store the attention keys and values of all previous tokens, eliminating the need to reprocess the

entire sequence during each decoding step. Instead, only the most recent token is passed through the model. However, this approach results in a different attention computation where the number of queries is 1 while the number of key-value pairs is still high ($N_q = 1$ and $N_{kv} \gg 1$). This leads to matrix-vector products for the multiplications with $K^T$ and $V$, making the attention during decoding memory-bound and not utilizing tensor cores.

## 3 METHOD

### 3.1 PROBLEM SETUP AND MOTIVATION

Consider a scenario where we have a shared context $Doc$ (e.g., a document) and $n$ questions $q_1, \ldots, q_n$ to answer based on this context. We want to generate answers $A = \{a_1, a_2, \ldots, a_n\}$ efficiently.

**Standard Approach with Shared Prefix.** The baseline processes each query independently, computing each answer as $a_i = \pi_{\mathsf{LLM}}(Doc, q_i)$, where $\pi_{\mathsf{LLM}}$ denotes the language model. For every query, the model performs a *prefill* pass or reuses part of the cache over the concatenated input $(Doc, q_i)$ and then executes a sequence of *incremental decode* steps to autoregressively generate the tokens of $a_i$. Answering $n$ queries requires repeating the computationally expensive generation stage $n$ times, which dominates overall runtime.

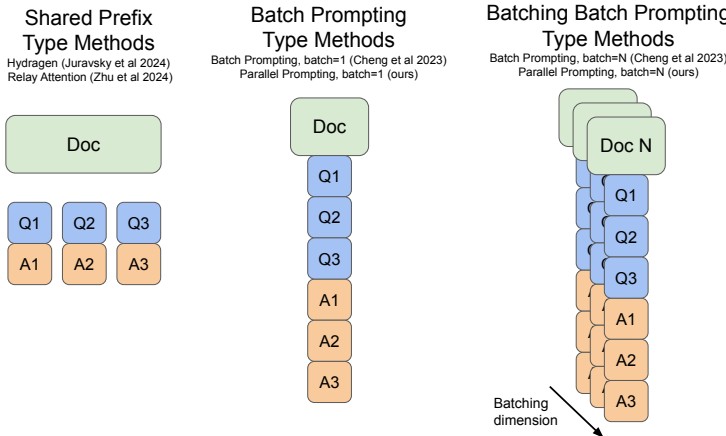

Figure 2: Methods for efficiently handling multiple prompts with a shared prefix. **Shared prefix type methods**, such as Hydragen and Relay Attention, batch together multiple questions and process them in parallel. **Batch prompting type methods** put multiple prompts together into one prompt, which can batch multiple documents together (**Batching Batch Prompting**).

**Batch Prompting Type Methods(SeqBatch)** (Cheng et al., 2023; Lin et al., 2024). A straightforward attempt to avoid redundant computation is to concatenate all queries into a single prompt and let the model generate a single long sequence containing all answers (see Figure 2, middle). While this approach amortizes the cost of encoding the shared context $Doc$, it introduces a *prompt interference* problem: due to the autoregressive nature of decoding, the model's hidden state at step $i$ contains all previously generated tokens. Consequently, the answer for query $i$ becomes implicitly conditioned on other questions and earlier answers, and the resulting outputs are no longer independent. This entanglement often degrades answer quality.

**Our Approach (Parallel Prompting).** We propose a method that generates all $n$ answers in parallel while ensuring that each answer remains conditioned only on its own query and the shared context. The central idea is to apply query-specific attention masks during both the prefill and decoding stages (see Figure 2, right), thereby isolating each question–answer flow while still enabling extensive sharing of computation. This yields three key advantages:

1. *Shared-context prefill:* the computationally intensive encoding of $Doc$ is executed once and reused for all queries;

2. *Parallel decoding:* at every generation step, the model produces multiple next tokens simultaneously for each query in a single batched forward pass;

3. *Independence of answers:* attention masking prevents cross-query information leakage, ensuring that the generation of $a_i$ depends solely on $(Doc, q_i)$.

Together, these mechanisms substantially reduce computation while preserving the independence and quality of the generated answers.

Our method integrates seamlessly with the batching technique. By batching texts with multiple unique documents and corresponding questions, efficiency can be improved further. Parallel generation with batching provides two distinct advantages: firstly, inference throughput is further amplified by batching with multiple unique prefix documents; secondly, it enables the balancing of batch size and sequence length for model input, optimizing overall performance.

## 3.2 PARALLEL GENERATION WITH PROMPT-WISE INDEPENDENT ENCODING

Our method operates in two stages: **Prefill** and **Parallel Decode**.

**Prefill Stage.** We concatenate all queries into a single input sequence and encode them jointly together with the shared context). To avoid any form of cross-query interference, we construct a query-specific attention mask (see Figure 1, right) that ensures each query token attends only to the shared context and to its own query tokens. This masking scheme is related to prepacking (Zhao et al., 2024), but here we extend it to support multiple independent decoding streams simultaneously. To preserve positional consistency, tokens for each query are assigned disjoint position indices immediately following the shared-context sequence. If the shared context has been previously prefetched, we directly reuse its KV-cache, thereby avoiding redundant prefill computation.

**Parallel Decoding Stage.** During autoregressive generation, we replace the standard one-token-per-query decoding pattern with a fully *parallel* decoding scheme. The SeqBatch method processes all documents and questions sequentially within a single batch. In contrast, the parallel generation method employs efficient matrix operations to process multiple documents and questions simultaneously, significantly accelerating the generation process by leveraging parallel computation capabilities. In each forward pass, the model generates $n$ tokens simultaneously (see Figure 1, right). This transforms attention operations from a sequence of memory-bound matrix–vector products into a single compute-bound matrix–matrix multiplication, resulting in significantly higher GPU utilization. The attention masks defined during prefill are reused at every decoding step and expanded following the same structural pattern, guaranteeing strict separation of decoding streams throughout generation.

The full algorithmic description is provided in Algorithm 1.

Since all questions are independent conditioned on the shared context, their answer distributions can be computed simultaneously. To support this, we allow the model to generate $N$ next-token logits in a single forward pass, which corresponds to constructing a query matrix $Q$ of shape $N \times d$ in the attention module.

During decoding, our method generates tokens for all questions in parallel. In each forward step, the model extends the sequence with $N$ new tokens—one per question. To maintain positional correctness, we track the final prefix position of each stream and increment the corresponding positional index before appending new tokens. After each step, the newly generated tokens are inserted into their respective query streams, and the attention masks are updated according to the fixed pattern defined during prefill. Because the mask structure is predetermined, only lightweight incremental updates are required.

## 3.3 THEORETICAL ANALYSIS

In this section, we present a theoretical analysis of the parallel prompting method, focusing on its efficiency gains in LLM inference. We begin by discussing the implications of Amdahl's Law in the context of parallel algorithms, followed by an examination of the speedup and throughput improvements achieved through our approach.

---

**Algorithm 1** Parallel_Batch_Prompting: Parallel Prompt Generation with Shared-Prefix Cache

---

**Require:** Shared prefix `Doc`, unique suffix set `Q_all`, batch size `N`, parallel size `P`, language model $\pi_{\text{LLM}}$

**Ensure:** List of generated answers
1: Optional: cache ← PRECOMPUTE($\pi_{\text{LLM}}$, Doc)                    ▷ Prefill KV-cache for shared prefix
2: i ← 0
3: N_p ← N / P                                                            ▷ Samples per parallel group
4: **while** i < |Q_all| **do**
5:     Q_n ← Q_all[i : i + N]
6:     Q_np ← PARALLIZEINTERLEAVE(Q_n, P)
7:     prompts ← PREPAREINPUT(Doc, Q_np, N_p)
8:     masks ← PREPAREMASK(prompts)
9:     answers, output_mask ← PARALLELGENERATE($\pi_{\text{LLM}}$, prompts, masks, P, cache)
10:     **for** n = 1 **to** N_p **do**
11:         **for** p = 1 **to** P **do**
12:             final_answer.append(DECODE(answers[n, p], output_mask[n, p]))
13:         **end for**
14:     **end for**
15:     i ← i + N
16: **end while**
17: **return** final_answer
18:
19: **function** PARALLELGENERATE($\pi_{\text{LLM}}$, prompts, masks, P, cache)
20:     finished ← False
21:     input_ids ← TOKENIZE(prompts)
22:     **while not** finished **do**
23:         outputs ← $\pi_{\text{LLM}}$.FORWARD(input_ids, masks, cache)
24:         logits ← outputs[:, -P:]                          ▷ Outputs P logits on sequence dimension
25:         next_tokens ← SAMPLE(logits)
26:         input_ids ← CONCAT(input_ids, next_tokens)
27:         **if** STOPPINGCRITERIA(input_ids) **then**
28:             finished ← True
29:         **else**
30:             masks ← UPDATEPARALLELMASK(input_ids, P)
31:         **end if**
32:     **end while**
33:     **return** input_ids, masks
34: **end function**
35:
36: **function** PRECOMPUTE($\pi_{\text{LLM}}$, Doc)
37:     kv_cache ← $\pi_{\text{LLM}}$.FORWARD(Doc)
38:     **return** kv_cache
39: **end function**

---

Amdahl's Law provides a theoretical framework for understanding the potential speedup of a task when a portion of it is parallelized. It is defined as:

$$S(N) = \frac{1}{(1 - p) + \frac{p}{N}} \tag{2}$$

where $S(N)$ is the speedup with $N$ processors, $p$ is the fraction of the task that can be parallelized, $1 - p$ is the fraction that remains serial. This law highlights that the overall speedup is limited by the serial portion of the task. As $N$ increases, the speedup approaches $\frac{1}{1-p}$, indicating diminishing returns if $p$ is not close to 1.

In the context of LLM inference, traditional methods process each query sequentially, leading to inefficiencies due to the serial nature of prompt processing. Our proposed method introduces parallel prompting, allowing multiple queries to be processed simultaneously. This approach effectively

maximizes throughput and reduces the time of the LLM's inference task. We measure throughput as queries (prompts) processed (a full output completion is generated) per unit time.

**Theorem 1** (Amdahl's Law for Inference Throughput Improvement). *The throughput improvement $\Delta$ (tasks processed per unit time above baseline) from using $N$-way parallel inference is:*

$$\Delta = \frac{N \cdot S(N) - 1}{T_{\text{seq}}} \tag{3}$$

*See proofs and further details in Equation A.1.*

**Proposition 2.** *Consider inference on $N$ independent queries using (a) standard batch processing and (b) parallel prompting (packing all queries as independent subsequences in a single sequence with attention masking).*

*Let $T_{batch} = T_{setup} + N \cdot T_{\text{MV}}$ be the wall-time for a batch (with matrix-vector attention), and $T_{parallel} = T_{setup} + T_{\text{MM}}$ for parallel prompting (with matrix-matrix attention). Then, the respective throughput values are:*

$$\text{Throughput}_{batch} = \frac{N}{T_{batch}}, \qquad \text{Throughput}_{parallel} = \frac{N}{T_{parallel}} \tag{4}$$

*and*

$$\frac{\text{Throughput}_{parallel}}{\text{Throughput}_{batch}} = \frac{T_{batch}}{T_{parallel}} = \frac{T_{setup} + NT_{\text{MV}}}{T_{setup} + T_{\text{MM}}} \tag{5}$$

*where $T_{\text{MV}}$ is per-query wall-time for the matrix-vector attentions, and $T_{\text{MM}}$ is wall-time for the matrix-matrix product in the attention.*

*In practical settings, due to the efficiency of matrix multiplications on a GPU, $T_{\text{MM}} \approx T_{\text{MV}}$. If $T_{setup} \ll T_{\text{MM}}$, then $\text{Throughput}_{parallel}$ is up to $N\times$ that of standard batching.*

While the theoretical analysis suggests significant improvements, practical factors such as communication overhead, memory bandwidth constraints, and synchronization costs can impact actual performance. It is essential to consider these factors when implementing parallel prompting to ensure that the theoretical gains translate into real-world efficiency.

### 3.4 THROUGHPUT MAXIMIZATION BY BALANCING ATTENTION PARALLELISM AND BATCH SIZE

The use of batching is a crucial technique to enhance throughput in LLM inference. Through batched decoding, each forward pass of the model processes the latest token from multiple sequences concurrently rather than just one. This approach amplifies the arithmetic intensity of transformer components, such as the multilayer perceptron (MLP) blocks, and facilitates the use of hardware-friendly matrix multiplications.

However, the computation intensity of attention does not inherently benefit from batching, as each sequence possesses its distinct key and value matrix. Consequently, while other model components can leverage tensor cores during batched decoding, attention is required to be computed using numerous independent matrix-vector products. Our parallel generation technique aims to address this by enhancing the computation intensity of attention.

**Proposition 3** (Throughput Maximization). *Let $P$ be the parallel size (number of independent queries packed into a sequence for matrix-matrix attention), $B$ the batch size (number of such sequences processed in parallel), and $P \cdot B \leq S^*$ a hardware resource constraint (e.g., total token capacity).*

*Let $T_{\text{attn}}(P)$ denote the attention computation cost (function of $P$), and $T_{\text{mlp}}(B)$ denote the MLP/other backend (function of $B$).*

*Then, the throughput (queries per unit time) satisfies:*

$$\text{Throughput}(P, B) = \frac{P \cdot B}{T_{\text{attn}}(P) + T_{\text{mlp}}(B)} \tag{6}$$

*and maximal throughput is achieved at*

$$(P^*, B^*) = \arg\max_{P \cdot B \leq S^*} \frac{P \cdot B}{T_{\text{attn}}(P) + T_{\text{mlp}}(B)} \tag{7}$$

*where $T_{\text{attn}}(P)$ generally improves with $P$ up to a hardware limit (then degrades), and $T_{\text{mlp}}(B)$ improves with $B$ up to a limit.*

The maximizing pair $(P^*, B^*)$ is found by balancing optimal matrix-matrix utilization for attention and optimal batch size for MLP efficiency. The throughput function is quasi-concave in $(P, B)$ under natural hardware scaling assumptions for transformer kernels. The theoretical maximum exists at an interior point determined by hardware and model specifics, and is not achieved by maximizing either $P$ or $B$ alone.

## 4 EXPERIMENTS

We evaluate our method through two complementary sets of experiments: (1) controlled scaling studies on small and medium-sized models using synthetic data, and (2) a downstream task evaluation on reading comprehension datasets using Llama 3–8B. This combination enables both fine-grained analysis of computational behavior and validation on a realistic application. All experiments are conducted on a single NVIDIA A100-80GB GPU using PyTorch implementations built on the HuggingFace architecture (Wolf et al., 2020). Additional implementation details are provided in Appendix B.

### 4.1 SCALING EXPERIMENTS

**Setup.** Following Juravsky et al. (2024), we construct synthetic datasets with varying document lengths, numbers of unique documents, and numbers of queries. Document content is drawn from a subset of *War and Peace* (Tolstoy, 1869), with procedurally generated sentences added for greater length diversity. We perform all scaling studies on CodeLlama-7B-Instruct (Rozière et al., 2024), Sheared-LLaMA-1.3B (Xia et al., 2024), and LLaMA-160M (Miao et al., 2023) to enable controlled analysis under constrained compute.

**Memory Constraints and Throughput Under Increasing Context Length.** We first examine memory usage and throughput as the number of queries and the shared-context length increase. Figure 3 summarizes the results. Several baselines (e.g., HuggingFace with DynamicCache, Hydragen) encounter out-of-memory failures at high query counts, whereas our method remains stable. As shown in the right panel of Figure 3, throughput decreases with longer prefixes for all methods, but our parallel prompting consistently achieves higher throughput without sacrificing generation quality. A full breakdown of memory measurements across all conditions appears in Table 6 and Table 5 in the Appendix.

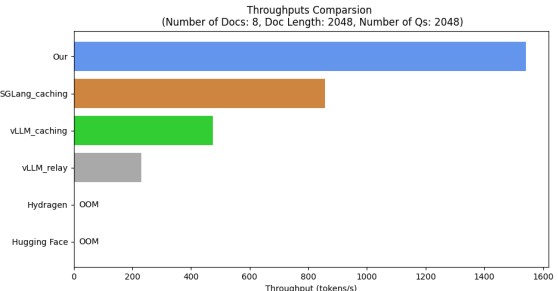 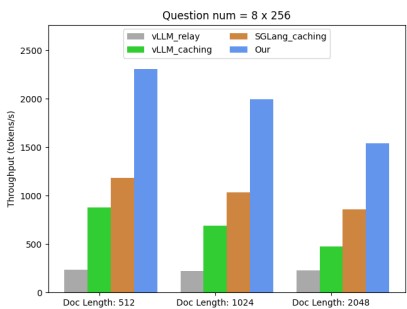

Figure 3: **Left:** Memory usage for multiple prefix-sharing methods under increasing numbers of queries with CodeLlama-7B-Instruct on an A100 GPU. **Right:** Throughput comparison for CodeLlama-7B-Instruct on an A100 GPU as the shared-context length increases. We fix 256 total queries, 8 unique documents, a query length of 12, and generate 5 tokens per query.

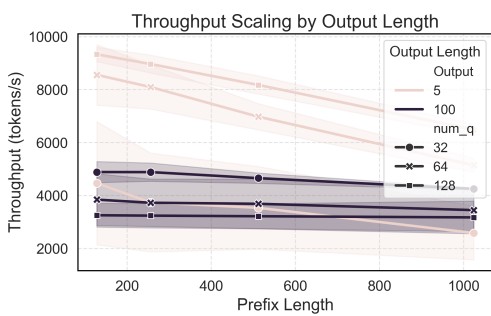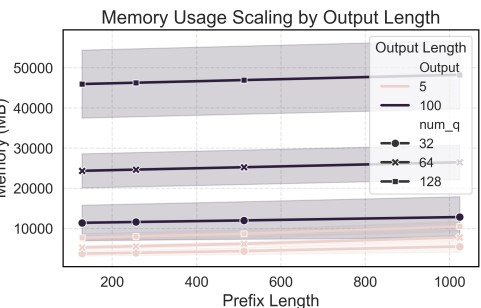

Figure 4: **Left:** Throughput (tokens/sec) as a function of prefix and output lengths with CodeLlama-7B-Instruct on an A100 GPU; lines correspond to different output lengths, markers denote the number of queries. **Right:** GPU memory usage with varying prefix and output lengths. Results shown for 4 documents and 32 queries.

**Scaling with Output Length.** To further isolate computational factors, we study performance as a function of generated output length. Figure 4 reports both throughput and GPU memory usage across varying prefix and output lengths. Longer prefixes and outputs impose higher computational load, but our method maintains efficiency and stable scaling.

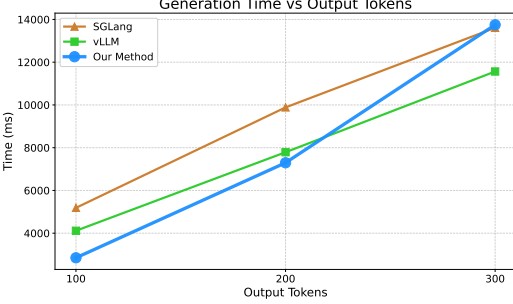

Figure 5: Comparison of generation time versus output tokens for our method, vLLM and SGlang with CodeLlama-7B-Instruct on an A100 GPU. As the number of output tokens increases, both methods require more time; however, our method consistently achieves lower generation time for shorter outputs and remains competitive as the output length grows. The blue line represents our method, while the light green line represents vLLM and the orange line represents SGLang, both evaluated with 4 documents and 32 questions per batch.

We also conduct experiments varying output length up to 300 tokens. Results on our syntactic dataset in Figure 5 show that Parallel Prompting consistently delivers throughput gains over the vLLM method up to approximately 200 output tokens per question. As an example, for four unique documents with $4 \times 32$ questions, our method required 7,295 milliseconds (throughput $\approx 3,500$ tokens/sec), while the vLLM method takes 7,605 milliseconds (throughput $\approx 3,360$ tokens/sec). When the output length exceeds 200 tokens, vLLM may offer a greater advantage.

**Batch Size vs. Parallel Size.** We next analyze how throughput depends jointly on batch size and parallel size. Intuitively, increasing parallel size improves efficiency up to a point, after which larger batch sizes provide better arithmetic intensity. Figure 6 (left and middle) illustrates that the optimal throughput is achieved by balancing these two factors. Our preliminary results suggest that longer prefixes prefer larger parallel size, as also visible in Figure 6 (Left). A detailed numerical comparison for 1B and 7B models appears in Table 4 in the Appendix. However, due to limited resources, we were unable to perform a comprehensive sweep across many model sizes and hardware settings.

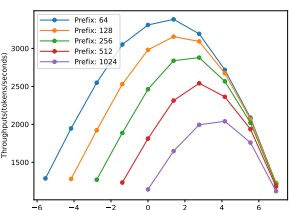 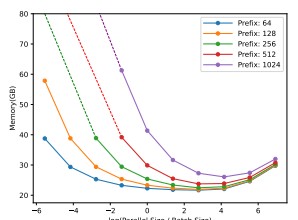 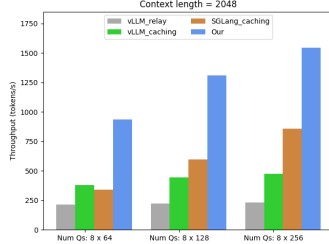

Figure 6: **Left:** Throughput comparison for 1024 queries across multiple document settings with CodeLlama-7B-Instruct on an A100 GPU. The X-axis represents the logarithm of the ratio between the parallel size and the batch size. This metric is used to show that these two parameters must be balanced to achieve maximum inference throughput. **Middle:** GPU memory usage for the same settings. **Right:** Throughput under long-context inference. Notation such as 8 × 64 means there are 8 unique documents, and each document has 64 associated questions (total = 512 questions).

## 4.2 CASE STUDY: QUESTION ANSWERING PERFORMANCE

We evaluate our method on downstream reading comprehension tasks to assess end-to-end impact on both quality and generation speed. We use Llama 3–8B (Grattafiori et al., 2024) and measure F1 scores (standard for QA) on SQuAD (Rajpurkar et al., 2016), QuAC (Choi et al., 2018), and DROP (Dua et al., 2019).

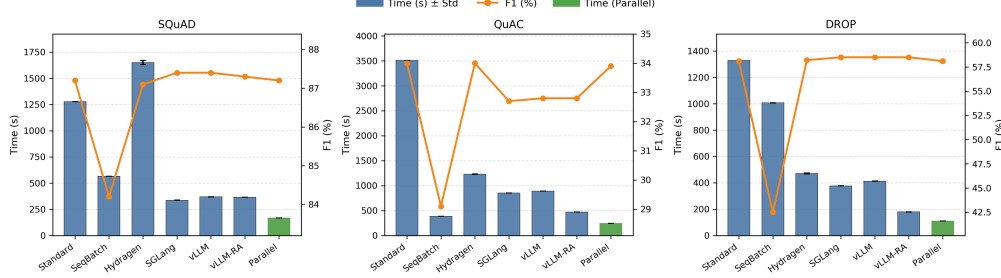

Figure 7: Comparison of generation time and F1 performance across prompting methods using Llama 3–8B on an A100 GPU. Reported results are averaged over five runs.

As shown in Figure 7, our parallel prompting achieves substantially lower latency compared to standard prompting, sequential batching, Hydragen, SGLang, vLLM (with and without relay attention), while maintaining equivalent answer quality across all datasets.

## 5 RELATED WORK

Recent advancements in language modeling have delved into the prediction of multiple tokens simultaneously to enhance both efficiency and performance. Notable works such as (Miao et al., 2024; Leviathan et al., 2023; Wu et al., 2024) focus on speculative decoding methods, where potential future sequences are built and verified to expedite inference. Similarly, (Gloeckle et al., 2024) and (Cai et al., 2024) propose predicting multiple future tokens using different output heads, thereby speeding up the inference process. Efforts to increase throughput in LLM inference have led to various innovative techniques aimed at optimizing GPU utilization and improving throughput. (Dao et al., 2022) and (Sheng et al., 2023) aim to improve memory usage efficiency, enabling higher throughput in generative inference tasks. (Jin et al., 2023) schedules prompts based on estimated output sequence lengths to optimize GPU usage. (Gim et al., 2024) proposes reusing precomputed caches in a predefined schema to reduce latency. (Sun et al., 2024) applies dynamic sparse KV caching in decoding to accelerate long sequence generation. Efficient prompting techniques could also increase the throughput of LLM.(Cheng et al., 2023) groups multiple questions in a single prompt,

though it will lead to performance degradation when the number of questions increases. (Zhao et al., 2024) enhances throughput during the prefilling stage by prepacking data. (Ning et al., 2024) uses the skeleton of the answer to batch-generate the final answer. To avoid the KV cache duplication, existing work (Kwon et al., 2023) vLLM uses its PagedAttention and paged memory management to point multiple identical input prompts to only one physical block across multiple queries. Also, (Juravsky et al., 2024) proposes a decomposition of attention computation of shared prefixes and unique suffixes. (Lu et al., 2024) increases efficiency by sharing cache in the encoder-decoder model for decomposable tasks. Compared with the above methods, our work introduces a novel inference technique that allows LLMs to leverage GPU parallel capacity to improve inference throughput and memory utilization without degrading reasoning performance.

## 6 CONCLUSION

We introduce an efficient parallel prompting method for decoding prompt queries in parallel. We conduct experiments with multiple downstream datasets, constructed synthetic data, and show our method achieves improvements in throughput and computational resource management, offering a robust solution for different tasks in LLMs.

## LIMITATIONS

**Skewed Generation Lengths**   Our method achieves the highest throughput gains when suffix lengths are similar, and performance may degrade when generation lengths are highly skewed during decoding. To mitigate this, we propose several practical strategies: In cases where generation lengths become highly unbalanced, the system can fall back to standard inference. In real-world applications, expected output length can often be heuristically estimated based on properties such as question and context length. This enables grouping questions with similar expected output lengths, minimizing skew. More advanced solutions, such as dynamic batching (e.g., as introduced in Verl), could be adopted to support streaming scenarios and further optimize batching efficiency.

**Prompt-Agnostic Batching**   Our method's gains are largest when there is a clear shared-prefix structure and output lengths are short to moderate. As the length of unique suffixes increases, the benefit of parallel generation diminishes, since more computation must be performed individually for each query. For very long outputs, prompt-agnostic batching (such as vLLM's default scheduling) may outperform our approach. We recommend a hybrid scheduling policy in production, using Parallel Prompting for workloads with substantial shared context and prompt-agnostic batching for others. This method is designed to complement, not replace, existing batching strategies.

## REPRODUCIBILITY STATEMENT

We have taken several steps to facilitate reproducibility. Assumptions and proofs for all theoretical claims are provided in Appendix [A], which states all conditions under which the results hold. Experimental settings—including datasets, preprocessing, model configurations, training schedules, hyperparameters, and evaluation protocols in Section Experiments. An anonymized, self-contained supplementary .zip archive includes source code and scripts to reproduce the main tables/figures and ablations. Known limitations, potential failure modes, and scope of applicability are discussed in Section Limitations. Any deviations from the default procedures or additional implementation notes are included in Appendix [B].

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

# A  APPENDIX

## A.1  PROOF OF THEOREM 1.

**Amdahl's Law for Inference Throughput Improvement**    The throughput improvement $\Delta$ (tasks processed per unit time above baseline) from using $N$-way parallel inference is:

$$\Delta = \frac{N \cdot S(N) - 1}{T_{\text{seq}}} \tag{8}$$

**Assumptions:**

- Each inference computation can be split into a parallelizable fraction and a sequential fraction.

- There are $N$ independent queries, each requiring $T_{\text{seq}}$ execution time if performed sequentially.

- There is no communication, scheduling, or parallelization overhead. Negligible coordination or resource contention.

- $N$ processors are available, and the parallel workload is divided equally among them. In parallel, independent $N$ queries are processed in time $T_{\text{par}}(N) = T_{\text{seq}}/S(N)$, where $S(N)$ is given by Amdahl's law Equation 2

*Proof of Theorem 1.*  The sequential throughput is $\frac{1}{T_{\text{seq}}}$. With parallel prompting, the time to process $N$ queries is $T_{\text{par}}(N)$, so the parallel throughput is $\frac{N}{T_{\text{par}}(N)}$. The improvement is:

$$\Delta = \frac{N}{T_{\text{par}}(N)} - \frac{1}{T_{\text{seq}}}$$

Assuming $T_{\text{par}}(N) = \frac{T_{\text{seq}}}{S(N)}$, we substitute to get:

$$\Delta = \frac{N}{\frac{T_{\text{seq}}}{S(N)}} - \frac{1}{T_{\text{seq}}} = \frac{N \cdot S(N)}{T_{\text{seq}}} - \frac{1}{T_{\text{seq}}} = \frac{N \cdot S(N) - 1}{T_{\text{seq}}}$$

$\square$

## A.2  ASSUMPTIONS OF PROPOSITION 2

Let $T_{\text{batch}} = T_{\text{setup}} + N \cdot T_{\text{MV}}$ be the wall-time for a batch (with matrix-vector attention), and $T_{\text{parallel}} = T_{\text{setup}} + T_{\text{MM}}$ for parallel prompting (with matrix-matrix attention). Then, the respective throughput values are:

$$\text{Throughput}_{\text{batch}} = \frac{N}{T_{\text{batch}}}, \qquad \text{Throughput}_{\text{parallel}} = \frac{N}{T_{\text{parallel}}} \tag{9}$$

and

$$\frac{\text{Throughput}_{\text{parallel}}}{\text{Throughput}_{\text{batch}}} = \frac{T_{\text{batch}}}{T_{\text{parallel}}} = \frac{T_{\text{setup}} + NT_{\text{MV}}}{T_{\text{setup}} + T_{\text{MM}}} \tag{10}$$

where $T_{\text{MV}}$ is per-query wall-time for the matrix-vector attentions, and $T_{\text{MM}}$ is wall-time for the matrix-matrix product in the attention.

**Assumptions:**

- The model and hardware support this masking and packing; $T_{\text{MV}}$ and $T_{\text{MM}}$ are measured compatibly.

- Time for setup is equal for standard batch processing and parallel prompting,

- $N$ is small enough to avoid exceeding hardware or memory limits for both methods.

### A.3 Assumptions of Proposition 3

**Throughput Maximization**  Let $P$ be the parallel size (number of independent queries packed into a sequence for matrix-matrix attention), $B$ the batch size (number of such sequences processed in parallel), and $P \cdot B \leq S^*$ a hardware resource constraint (e.g., total token capacity).

Let $T_{\text{attn}}(P)$ denote the attention computation cost (function of $P$), and $T_{\text{mlp}}(B)$ denote the MLP/other backend (function of $B$).

Then, the throughput (queries per unit time) satisfies:

$$\text{Throughput}(P, B) = \frac{P \cdot B}{T_{\text{attn}}(P) + T_{\text{mlp}}(B)} \tag{11}$$

and maximal throughput is achieved at

$$(P^*, B^*) = \arg\max_{P \cdot B \leq S^*} \frac{P \cdot B}{T_{\text{attn}}(P) + T_{\text{mlp}}(B)} \tag{12}$$

where $T_{\text{attn}}(P)$ generally improves with $P$ up to a hardware limit (then degrades), and $T_{\text{mlp}}(B)$ improves with $B$ up to a limit.

**Assumptions:**

- $P$ queries packed per prompt, $B$ prompts in a batch, $PB \leq S^*$ (resource or hardware constraint).
- Model/hardware supports this arrangement; $T_{\text{attn}}(P)$ and $T_{\text{mlp}}(B)$ are the attention/MLP module wall times.
- $T_{\text{attn}}(P), T_{\text{mlp}}(B)$ are nonincreasing (improve) up to hardware limits, then nonmonotone.

## B  Technical Appendices and Supplementary Material

The decision to use different models and datasets for the analytical and ablation studies, as compared to the main downstream task evaluations, is motivated by both practical and scientific considerations. Large models like Llama 3-8B are computationally intensive, making it challenging to run extensive ablation and scaling experiments across a wide range of parameters. By using smaller models and synthetic datasets for these studies, we are able to systematically vary key factors (such as batch size, prefix length, and number of queries) and isolate the effects of our method in a controlled environment. This approach enables us to provide deeper insights into the scaling laws, bottlenecks, and generalization of our method, while reserving the large-scale, real-world benchmarks for the main results. We believe this combination offers a comprehensive and rigorous evaluation of our approach.

Table 1: Comparison of generation time and performance for downstream tasks with different methods on average of five times with Llama 3 8B model on A100-80G. Std denotes the across-run standard deviation of the time. F1 is computed as the harmonic mean of precision and recall in extractive QA.

| Method | SQuAD | | | QuAC | | | DROP | | |
|--------|-------|-----|-------|------|-----|-------|------|-----|-------|
| | Times(s) | Std | F1(%) | Time(s) | Std | F1(%) | Time(s) | Std | F1(%) |
| Standard | 1277 | 0.08 | 87.2 | 3512 | 0.06 | 34.0 | 1330 | 0.08 | 58.1 |
| SeqBatch | 566 | 0.21 | 84.2 | 386 | 0.10 | 29.1 | 1007 | 0.41 | 42.5 |
| Hydragen | 1651 | 20.9 | 87.1 | 1230 | 6.74 | 34.0 | 471 | 3.85 | 58.2 |
| SGLang | 337 | 0.49 | 87.4 | 854 | 0.17 | 32.7 | 377 | 0.56 | 58.5 |
| vLLM | 369 | 0.46 | 87.4 | 889 | 0.57 | 32.8 | 413 | 0.44 | 58.5 |
| vLLM-RA | 365 | 0.21 | 87.3 | 469 | 0.15 | 32.8 | 179 | 0.51 | 58.5 |
| **Parallel** | 167 | 0.16 | 87.2 | 243 | 0.32 | 33.9 | 110 | 0.09 | 58.1 |

**Memory Usage**  The observed increase in memory usage for the Parallel method on datasets results from dynamically maximizing batch sizes during inference. Our approach allows processing more

examples in a fixed memory footprint, improving throughput. To validate this, we reduced the maximum allowed batch size during inference on QuAC and observed a significant drop in memory usage, while still demonstrating substantial speedup over the baseline with the maximum possible batch size. For transparency, Table 3 lists the results across different batch size settings with our method. This demonstrates that our method flexibly trades off memory and throughput by adjusting batch size, and can achieve substantial speedup even at lower memory footprints.

Table 2: Comparison of memory usage with different methods with Llama 3 8B model on A100-80G.

| Dataset | Method | Time(s) | Memory(GB) |
|---------|--------|---------|-----------|
| SQuAD | Standard | 590 | 55.7 |
| | Parallel | 168 | 48.6 |
| QuAC | Standard | 1799 | 55.0 |
| | Parallel | 352 | 33.1 |
| DROP | Standard | 654 | 54.3 |
| | Parallel | 111 | 36.1 |

Table 3: QuAC: Inference Time and Memory Usage for Different Batch Sizes (Parallel Method)

| Batch Size | Inference Time (s) | Memory (GB) |
|-----------|-------------------|-------------|
| Baseline | 1799 | 55.0 |
| 8 | 872 | 16.9 |
| 16 | 677 | 19.8 |
| 32 | 420 | 24.7 |
| 64 | 352 | 33.1 |
| 128 | 342 | 54.0 |

**Effect of the Number of Questions.** We sweep over the number of queries for fixed document and query lengths. Table 4 shows that throughput improves as the number of parallel queries increases, particularly for larger models. At small batch sizes, non-attention operations dominate, but at large query counts, attention over long prefixes becomes the bottleneck—precisely where our parallel decoding provides the largest gains.

Table 4: Throughput (tokens/sec) under different batch sizes for parallel generation with CodeLlama-1B and CodeLlama-7B when $doc\_len = 512$, $q\_len = 12$, and $ans\_len = 5$.

| Num_Questions | Batch Size | Throughput-1B | Throughput-7B |
|---------------|-----------|---------------|---------------|
| 128 | 1 | 4283 | 1931 |
| | 2 | 4625 | 1843 |
| | 4 | 3654 | 1468 |
| | 8 | 2850 | 1018 |
| 256 | 1 | 5911 | 2115 |
| | 2 | 6384 | 2250 |
| | 4 | 5748 | 2071 |
| | 8 | 4959 | 1615 |
| 512 | 1 | 5419 | 1850 |
| | 2 | 6845 | 2214 |
| | 4 | 7725 | 2382 |
| | 8 | 7181 | 2146 |

**Sequence Length vs. Computation Gains Trade-off.** Both theory and empirical results confirm that throughput increases with batch/parallel size up to a point—after which the computational

overhead of longer input sequences (from packed prompts) outweighs the matrix-matrix compute advantage. For example, on A100s, parallel sizes between 32 and 64 are optimal for typical workloads.

**Compatibility with Speculative Decoding.** Parallel Prompting (fanning out multiple suffixes at lock-step) is designed for simultaneous multi-query generation, while speculative decoding focuses on verifying a single sequence. These are distinct but potentially complementary: speculative decoding could be performed within each branch created by Parallel Prompting, or adapted to verify multiple shared-prefix continuations in parallel.

**Developer Overhead and Practical Adoption.** In many production stacks, the shared-prefix boundary is already explicit: for example, retrieval-augmented generation (RAG) pipelines concatenate retrieved context (prefix) with a question (suffix), and batched APIs naturally group queries under a common header or instruction. In these settings, enabling Parallel Prompting requires only providing: (1) the token span (or delimiter) for the shared prefix, and (2) a list of per-query suffixes. This makes practical adoption straightforward in most modern LLM serving pipelines.

**Memory Scaling Experiments** To systematically study memory and throughput scaling, we conducted experiments varying shared prefix length (128, 256, 512, 1024 tokens), output length (5 vs 100 tokens), number of unique prefixes (num_doc: 4 vs 8), and number of questions per prefix (num_q: 32, 64, 128). Our results reveal several key patterns: (1) Output length is the dominant driver of memory usage, followed by num_doc and context length, with num_q having a smaller but non-negligible effect. (2) Long outputs dominate memory via KV cache growth across all decode steps. (3) num_doc has a much larger impact when output is long, as a longer context is carried through every generated token. (4) Longer shared prefixes add memory, but the effect is modest compared to output length and num_doc, consistent with effective prefix sharing across the batch.

Table 5: Memory Usage (MB) and Throughput (tokens/s) for Output Length 100

| Prefix | num_doc | num_q | Memory (MB) | Throughput (tok/s) |
|--------|---------|-------|-------------|--------------------|
| 128 | 4 | 32 | 7031 | 4490 |
| 128 | 8 | 32 | 15814 | 5286 |
| 128 | 4 | 64 | 20104 | 4825 |
| 128 | 8 | 64 | 28617 | 2868 |
| 128 | 4 | 128 | 37509 | 3704 |
| 128 | 8 | 128 | 54429 | 2810 |
| 256 | 4 | 32 | 7131 | 4540 |
| 256 | 8 | 32 | 16109 | 5231 |
| 256 | 4 | 64 | 20399 | 4624 |
| 256 | 8 | 64 | 28927 | 2834 |
| 256 | 4 | 128 | 37829 | 3705 |
| 256 | 8 | 128 | 54761 | 2780 |
| 512 | 4 | 32 | 7333 | 4462 |
| 512 | 8 | 32 | 16689 | 4852 |
| 512 | 4 | 64 | 20968 | 4627 |
| 512 | 8 | 64 | 29545 | 2752 |
| 512 | 4 | 128 | 38472 | 3692 |
| 512 | 8 | 128 | 55433 | 2747 |
| 1024 | 4 | 32 | 7766 | 4289 |
| 1024 | 8 | 32 | 17932 | 4217 |
| 1024 | 4 | 64 | 22174 | 4262 |
| 1024 | 8 | 64 | 30787 | 2639 |
| 1024 | 4 | 128 | 39751 | 3792 |
| 1024 | 8 | 128 | 56787 | 2559 |

**Effect of Model Size** The performance of LLM's generation can be affected by various factors such as number of queries, batch size and the length of prefixes. We also run experiments with various

Table 6: Memory Usage (MB) and Throughput (tokens/s) for Output Length 5 tokens

| Prefix_Length | Num_Documents | Num_Questions | Memory (MB) | Throughput (tok/s) |
|---|---|---|---|---|
| 128 | 4 | 32 | 3187 | 2144 |
| 128 | 8 | 32 | 4324 | 6794 |
| 128 | 4 | 64 | 4798 | 7412 |
| 128 | 8 | 64 | 5767 | 9700 |
| 128 | 4 | 128 | 6735 | 9060 |
| 128 | 8 | 128 | 8724 | 9602 |
| 256 | 4 | 32 | 3290 | 1875 |
| 256 | 8 | 32 | 4622 | 5605 |
| 256 | 4 | 64 | 5095 | 7264 |
| 256 | 8 | 64 | 6073 | 8928 |
| 256 | 4 | 128 | 7039 | 8627 |
| 256 | 8 | 128 | 9041 | 9304 |
| 512 | 4 | 32 | 3512 | 1976 |
| 512 | 8 | 32 | 5260 | 5098 |
| 512 | 4 | 64 | 5687 | 6479 |
| 512 | 8 | 64 | 6684 | 7472 |
| 512 | 4 | 128 | 7671 | 7831 |
| 512 | 8 | 128 | 9727 | 8520 |
| 1024 | 4 | 32 | 4143 | 1573 |
| 1024 | 8 | 32 | 6906 | 3601 |
| 1024 | 4 | 64 | 7135 | 4882 |
| 1024 | 8 | 64 | 8426 | 5404 |
| 1024 | 4 | 128 | 9261 | 6282 |
| 1024 | 8 | 128 | 11751 | 6689 |

configurations with CodeLlama-7b-Inst (Rozière et al., 2024) and Sheared-LLaMA-1.3B (Xia et al., 2024) since different model sizes could also affect generation performance. See Table 7 for results.

Table 7: Comparing the throughput using parallel Batching with 7B and 1B Llama model with different lengths of doc length when $q\_len = 12 \| q\_num = 128 \| ans\_len = 5$ and the number of unique doc content equals 8. As the content length increases, the degradation of throughput performance becomes severe.

| $doc\_len$ | $Throughput(1B)(tokens/second)$ | $Throughput(7B)(tokens/second)$ |
|---|---|---|
| 256 | 9512 | 2750 |
| 512 | 8199 | 2430 |
| 1024 | 6591 | 1924 |

