# OpenReview forum: "Parallel Prompting: Fast LLM Inference for Shared-Context, Short-to-Moderate Output"
_ICLR.cc/2026/Conference — Submitted to ICLR 2026_

### Official Review · Reviewer_Buei · 2025-10-26

**Soundness:** 3
**Presentation:** 3
**Contribution:** 3
**Rating:** 6
**Confidence:** 2

**Summary:**

This work focuses a practical question: how to efficiently decode multiple queries that share a common prefix. The Parallel Prompting finds that the inference throughput requires a careful balance between attention parallelism and batch size.

**Strengths:**

* This work try to improve the efficiency of generation that shares a common prefix in large language models (LLMs). For example, the current LLM usually has a system prompt.
* The Figure 1 presents that there is an optimal point between the parallel size and batch size. This is the strength of the motivation of this work:.
* This work significantly improves the generation thoughtoutput, as presented in Figure 1.
* This work sufficient discuss the effect of query number and query length in Figure 3
* In Table 1, this work almost reduce half time cost than the vllm.

**Weaknesses:**

* Is the Parallel Size P the number of sub batch that are process? In Algorithm 1, why logits ← outputs[:, −P :], as the P should be at the batch size dimension?
* In Figure 1 middle, why the increase of log(parallel/batch_size) will lead to the memory cost decrease?
* Does this work propose a method to estimate the optimal P for the processing? How to determine the Parallel Size P?
* The method performance may not be good with long context.

**Questions:**

N/A

---

> ### Author Response · Authors · 2025-11-15
>
> Thank you for the thoughtful evaluation and for highlighting our strengths. Below, we address your questions and concerns in detail.
>
> 1.  Parallel Size P refers to the number of parallel decoding streams produced from the shared prefix. In Algorithm 1, the tensor layout places the P parallel streams in the sequence dimension, not the batch dimension. Therefore, outputs[:, −P :] extracts the last P positions of the sequence (the positions corresponding to the parallel branches), which is correct under this layout. We will revise the text to explicitly state the tensor shapes to avoid confusion.
>
> 2. In Figure 1 (Middle), why does increasing log(parallel/batch_size) decrease memory cost?
> Increasing log(parallel/batch_size) corresponds to increasing P (parallel size) and decreasing B (batch size). This reduces the number of independent KV caches stored per generation request and increases the proportion of shared KV states. The result is more efficient KV-cache usage during decoding and, therefore, lower memory cost.
>
> 3. Does the paper propose a method to estimate the optimal P? How should the Parallel Size P be determined?
> We present empirical observations rather than a full predictive model. Our preliminary results suggest that longer prefixes prefer larger P, as also visible in Figure 1 (Left). However, due to limited hardware, we were unable to perform a comprehensive sweep across many model sizes and contexts. We will clarify this limitation in the revision and describe the observed trend more explicitly.
>
> 4. Concern with long context lengths:
> Long contexts do not become a performance bottleneck for our method as long as the generation length is moderate. Because our approach avoids large prefill-time allocations and uses shared KV caches efficiently, long prefixes do not degrade performance.
>
> Once again, we thank the reviewer for their insightful comments.

---

### Official Review · Reviewer_aDJT · 2025-10-30

**Soundness:** 1
**Presentation:** 1
**Contribution:** 1
**Rating:** 2
**Confidence:** 3

**Summary:**

This paper introduces Parallel Prompting to efficiently decode multiple queries that share a common prefix in LLMs. This method introduces a method for efficiently generating answers to multiple questions in parallel by independently encoding prompts and leveraging shared context in large language models. Experimental results show that Parallel can improve end-to-end Llama3-8B latency by up to 4× against competitive baselines, without compromising output quality.

**Strengths:**

Experimental results show its effectiveness.

**Weaknesses:**

1. The writing is poor in Lines 110 - 151. This makes the paper very difficult to understand.

2. The paper is very difficult to understand in terms of what was actually done and why the method is effective. Its algorithm is highly non-intuitive.

3. The second and third paragraphs of the introduction are written very poorly. The author hasn’t even figured out how to organize their own work.

4. This approach requires a longer context. Wouldn't that lead to increased computational costs?

5. Why does SeqBatch take less time on the QuAC dataset?

6. The placement of figures and tables in the experimental section of this paper needs to be rearranged.

7. In Line 110, Suppose we have a context C and N sentence queries -> Suppose we have a context C and n sentence queries7

**Questions:**

See weaknesses

---

> ### Author Response · Authors · 2025-11-15
>
> Thank you for your thoughtful and detailed feedback. We will revise these sections to improve organization, clarify motivation, and present the algorithm in a more intuitive and structured manner, including the addition of high-level explanations before technical details. We will add clarification on the computational implications of longer contexts, explaining that although longer contexts generally increase cost, our method amortizes this overhead through shared caching. Regarding the QuAC dataset results, we now clearly explain that SeqBatch—used as one of our baselines—requires encoding the shared context only once, which allows it to run faster than strong baselines like vLLM and SGLang. We appreciate your comments, which have encouraged us to improve the clarity and presentation of the paper.

---

> > ### Comment · Reviewer_aDJT · 2025-11-15
> >
> > Could you please submit a new PDF so that I can review it directly? I remember that ICLR 2025 allowed this, but I’m not sure whether it is still allowed this year.

---

> ### Author Response · Authors · 2025-11-22
>
> Thanks for your constructive comments and suggestions. We have carefully revised the manuscript.  The main revisions and enhancements can be summarized as follows:
>
> - Improved presentation (Sec. 1, Sec. 3.1, Sec. 3.2).
>
> - Rearranged Results in the experimental section (Sec. 4).
>
> - Results on downstream with figures (Sec. 4.2).
>
> - Adding SGLang results with longer output tokens (Fig. 5)
>
> We hope the revised version meets your expectations and look forward to your continued feedback.

---

### Official Review · Reviewer_KAvc · 2025-10-31

**Soundness:** 3
**Presentation:** 3
**Contribution:** 2
**Rating:** 6
**Confidence:** 3

**Summary:**

This paper introduces Parallel Prompting, a novel method for efficiently decoding multiple queries that share a common prefix in large language models. By leveraging parallel processing and matrix-matrix operations, the method significantly improves inference throughput while maintaining output quality. The authors provide theoretical grounding and extensive experiments demonstrating its effectiveness, particularly for short-to-moderate output lengths with high prefix overlap.

**Strengths:**

1. Novelty: The proposed method effectively addresses inference inefficiencies in shared-prefix scenarios.

2. Theoretical Foundation: A solid analysis based on Amdahl’s Law and hardware constraints explains the trade-off between parallel size and batch size.

3. Comprehensive Experiments: Systematic evaluations across multiple models and datasets show improvements in throughput, memory usage, and output quality.

**Weaknesses:**

1. Advantage Diminishes and Reverses with Longer Outputs: This is the most significant limitation. As shown in this paper, the generation time of the proposed method begins to exceed that of vLLM when the output length per query exceeds approximately 200 tokens. Therefore, this method is applicable to a quite limited range of tasks.

2. System Complexity and Scheduling Overhead: To achieve optimal performance, the method necessitates a sophisticated scheduler to dynamically balance parallel size (P) and batch size (B). This scheduler must make real-time decisions based on hardware specs, model size, prefix length, output length, etc. The computational and logical complexity of this scheduling itself, compared to the relatively "dumb" but simple scheduler in vLLM, represents an additional engineering and runtime cost that the paper does not evaluate.

**Questions:**

1. In extreme cases, such as when the prefix length reaches tens of thousands of tokens and the number of queries is also large, could the initial prefill stage and memory usage of the method become a new bottleneck? How would it perform compared to other methods?

2. Given the core weakness identified, the paper suggests a hybrid scheduling policy in production. Could you elaborate on the design principles of such a hybrid scheduler? For instance, would the decision to use Parallel Prompting or fall back to a dynamic batching method (like vLLM's default) be based on the estimated output length or the real-time observed generation length?

3. The paper identifies that maximizing throughput requires balancing the parallel size (P) and batch size (B), and finds the optimal (P, B) through experimental search. However, for production systems, the cost of this search itself can be high. Could the authors propose a low-cost method or a rule of thumb (e.g., building a predictive model based on key factors like model size, prefix length, available memory, etc.) to efficiently determine or dynamically adapt near-optimal P and B values, rather than relying on expensive grid search?

---

> ### Author Response · Authors · 2025-11-15
>
> Thank you for your constructive review and for recognizing both the strengths and the specific areas for improvement in our method. We address each point below:
>
> 1. Prefix length and prefill/memory bottlenecks:
> Our method does not preallocate large memory regions nor require expensive precomputation during the prefill stage, so extremely long prefixes (tens of thousands of tokens) do not create a new systemic prefill bottleneck for our approach. In practice, our memory footprint is smaller than HuggingFace’s DynamicCache and Hydragen-style approaches because we avoid extensive KV pre-allocation. LLM-serving systems that use dynamic batching (e.g., vLLM, SGlang) group requests of differing lengths into variable-sized GPU batches, so their instantaneous memory usage depends on the dynamic batch composition and cannot be directly compared to a fixed-allocation scheme. A fairer comparison is in throughput under matched latency or memory caps; we therefore focus comparisons on throughput while reporting observed memory behavior.
> 2. Design principles for a hybrid scheduler (Parallel Prompting with dynamic batching):
> Classifier + policy architecture: the scheduler first classifies incoming requests (offline estimate or light-weight runtime probe) into categories such as short/interactive versus long/best-effort.
> Decision signals: use inexpensive signals (estimated output length from prompt heuristics, client-supplied max_tokens, historical average generation length for that client or workload, and current system load) to pick a decoding mode.
> - Two-mode operation:
> - - Parallel Prompting mode for short, latency-sensitive requests — prioritize low tail latency and high concurrency.
> - - Dynamic-batching mode (vLLM/SGlang-style) for long or throughput-oriented jobs — prioritize GPU utilization and cost efficiency.
> - Adaptive switching: allow switching mid-generation when a job crosses a threshold (e.g., after N generated tokens or when predicted remaining length grows); implement this by checkpointing the decoding state and migrating to the alternative decoder.
> 3. Rule-of-thumb: set parallel size:
> P is roughly proportional to the prefix length and inversely proportional to per-token KV cost. This captures the empirical observation that longer prefixes favor larger P (see Figure 1 (Left)).
> - Due to constrained hardware resources, we did not run a full systematic sweep across many model sizes and GPU types; we will clarify this limitation in the paper and release our scripts and microbenchmarks so others can reproduce and extend the search on different hardware.
>
> Once again, we thank the reviewer for their insightful comments.

---

### Official Review · Reviewer_XfJt · 2025-10-31

**Soundness:** 2
**Presentation:** 1
**Contribution:** 2
**Rating:** 4
**Confidence:** 2

**Summary:**

This paper proposes ParallelPrompting for efficiently serving many document - many questions scenarios.

**Strengths:**

ParallelPrompt can speed up overall serving throughput

**Weaknesses:**

I have much confusion about the figures and the main texts. Please review my questions.

**Questions:**

### Questions
What is the fundamental difference between RelayAttention + Prefix cached batched decode?

Figure 1 Left Center. X Label should be formatted formally. What do you mean by Parallel Size / Batch Size exactly?

Figure 1 Left Center. The legend does not appear to be formally formatted. What is Length(Doc)? Context length? Length of shared tokens? Number of documents in request?

Figure 1 Right. What is 8 x 64, 8x 128, 8x 256? Does it mean (document count ==) batch size and parallel question sizes? X label looks confusing.

Figure 2. Did you use huggingface's static cache? Why is it OOM?

Figure 4. Why no SGlang?

Figures should be PDF or SVG exported

Table 1. What is lossless, or not (vLLM and SGlang must be lossless, right?)

Table 3. num_q -> change to formal naming

### Formattings
Algorithm 1: Why are local variables not plain text? Font formatting should be textt or text

Typo line 320: inference.CodeLlama -> inference. [ ] CodeLlama

---

> ### Author Response · Authors · 2025-11-15
>
> We appreciate the reviewer’s feedback and have addressed each concern with clarification and additional results.
>
> 1. Fundamental difference between RelayAttention and Prefix-cached batched decode :
> RelayAttention with  Prefix-cached batched decoding (vllm-ra) cannot support batch inference with multiple system prompts where each has multiple questions, because it requires a fused operator capable of handling hybrid batches with multiple sharing groups—something that is technically challenging for their method.
>
>
> 2. Figure 1 (Left): X-axis meaning :
> The X-axis represents the logarithm of the ratio between the parallel size and the batch size. This metric is used to show that these two parameters must be balanced to achieve maximum inference throughput.
>
>
> 3. Figure 1 (Left): Legend clarification:
>  Length(Doc) refers to the number of shared tokens (i.e., the shared context). The number of unique documents equals the batch size.
>
>
> 4. Figure 1 (Right): Meaning of “8 × 64,” “8 × 128,” etc. :
>  Notation such as 8 × 64 means there are 8 unique documents, and each document has 64 associated questions (total = 512 questions).
>
>
> 5. Figure 2: HuggingFace cache and OOM:
>  We use HuggingFace’s DynamicCache, which can still lead to OOM issues in long-context generation because the KV cache consumes a large amount of memory, especially on GPUs.
>
>
> 6. Figure 4: Why no SGlang results:
>  We ran SGlang under the same settings. Its generation time was 5187 ms (100 tokens), 9885 ms (200 tokens), and 13609 ms (300 tokens). These results do not alter our claims: our method remains faster for short outputs and competitive as output length increases.
>
>
> 7. Figure format (PDF/SVG):
>  We will re-export and replace all figures with vector formats (PDF or SVG).
>
>
> 8. Table 1: Meaning of “lossless”:
>  “Lossless” refers to methods that preserve output quality. vLLM and SGlang are lossless; methods like SeqBatch may reduce generation fidelity.
>
>
> 9. Table 3: “num_q” naming:
>  We will replace num_q with a more formal and descriptive variable name.
>
> Thank you again for your valuable suggestions. We will make these new changes in the revised paper.

---

### Meta-Review · Area_Chair_eSGm · 2026-01-06

**Summary:**

The writing is not meeting the bar, as many reviewers found it hard to understand.

**Reviewer Concerns:**

The results from SGLang were added by authors. The OOM issue was not addressed.

**Reviewer Scores:**

The revised paper was not re-checked so if the improvement of presentation is acknowledged remains uknown.

---

### Decision · Program_Chairs · 2026-01-26

Reject